# Flexible Wireless Passive LC Pressure Sensor with Design Methodology and Cost-Effective Preparation

**DOI:** 10.3390/mi12080976

**Published:** 2021-08-18

**Authors:** Zhuqi Sun, Haoyu Fang, Baochun Xu, Lina Yang, Haoran Niu, Hongfei Wang, Da Chen, Yijian Liu, Zhuopeng Wang, Yanyan Wang, Qiuquan Guo

**Affiliations:** 1College of Electronics and Information Engineering, Shandong University of Science and Technology, Qingdao 266590, China; sunzhuqi2021@163.com (Z.S.); fanghy0820@163.com (H.F.); xbcno1@foxmail.com (B.X.); yln_lina@163.com (L.Y.); nhaoran2019@163.com (H.N.); sdkdwhf@163.com (H.W.); wzhuopeng@126.com (Z.W.); 2School of Optoelectronic Science and Engineering Collaborative Innovation Center of Suzhou Nano Science and Technology, Suzhou 215556, China; yywang@suda.edu.cn; 3Shenzhen Institute for Advanced Study, University of Electronics Science and Technology of China, Shenzhen 518110, China; guoqiuquan@126.com

**Keywords:** flexible LC pressure sensor, Ecoflex microcolumn, coplanar waveguide-fed monopole antenna, physical motion signal measurement

## Abstract

Continuous monitoring of physical motion, which can be successfully achieved via a wireless flexible wearable electronic device, is essential for people to ensure the appropriate level of exercise. Currently, most of the flexible LC pressure sensors have low sensitivity because of the high Young’s modulus of the dielectric properties (such as PDMS) and the inflexible polymer films (as the substrate of the sensors), which don’t have excellent stretchability to conform to arbitrarily curved and moving surfaces such as joints. In the LC sensing system, the metal rings, as the traditional readout device, are difficult to meet the needs of the portable readout device for the integrated and planar readout antenna. In order to improve the pressure sensitivity of the sensor, the Ecoflex microcolumn used as the dielectric of the capacitive pressure sensor was prepared by using a metal mold copying method. The Ecoflex elastomer substrates enhanced the levels of conformability, which offered improved capabilities to establish intimate contact with the curved and moving surfaces of the skin. The pressure was applied to the sensor by weights, and the resonance frequency curves of the sensor under different pressures were obtained by the readout device connected to the vector network analyzer. The experimental results show that resonant frequency decreases linearly with the increase of applied pressure in a range of 0–23,760 Pa with a high sensitivity of −2.2 MHz/KPa. We designed a coplanar waveguide-fed monopole antenna used to read the information of the LC sensor, which has the potential to be integrated with RF signal processing circuits as a portable readout device and a higher vertical readout distance (up to 4 cm) than the copper ring. The flexible LC pressure sensor can be attached to the skin conformally and is sensitive to limb bending and facial muscle movements. Therefore, it has the potential to be integrated as a body sensor network that can be used to monitor physical motion.

## 1. Introduction

As life expectancy and various health risks are simultaneously on the rise, people usually do physical exercises to maintain a healthy body. In order to reduce the risk of exercise injury, it is essential to continuously monitor limb motion magnitude for human health. Traditional rigid biomedical sensors have mechanical properties which are significantly different from those of the soft and curved surfaces of human tissues [1]. Flexible pressure sensors have been widely used in applications as wearable devices for monitoring human motion. But most sensors require wire connections for data transmission, which will greatly limit the users’ mobility and interfere with their daily lives [2,3,4,5]. For flexible wearable electronic equipment, the application of the wireless communication technique is significant to comprehensively evaluate the users’ health condition and ensure their comfort [6]. Wireless sensors are classified into two main technology branches, the active sensor and the passive sensor. The emergence of active wireless sensors has solved the problem of wiring connection, but the existence of batteries has brought a series of problems [7,8]. The batteries usually take up a large volume and must be recharged or replaced regularly. Therefore, supplying self-sustainable power is important for the wireless system as it can eliminate the need to periodically replace the power source [9]. So far, a lot of works have been devoted to designing wireless systems based on commercially available wireless devices such as Bluetooth and near-field communication (NFC) chips, which can be connected to an external readout system for monitoring human health [10,11,12,13,14,15]. However, considering the effect of rigidity and shape factors, when the device is attached to human skin, the discomfort inevitably limits the freedom of user motion. Hence, wireless devices should be endowed with stretchability and flexibility, similar to human skin, which can be achieved by using an LC resonator based on the electromagnetic coupling principle.

In recent years, flexible wireless passive LC pressure sensors have been a hot research field. They provided a convenient solution that did not require power supplies or rigid chips, thereby making it well suited for wearable devices. Usually, the planar inductance is not only the basic unit that constitutes the LC electric resonant circuit but also acts as the antenna that achieves the transmission function of energy and data with the readout system. Furthermore, the capacitance acts like a pressure-sensitive unit that changes in response to different external pressures. Therefore, flexible LC pressure sensors have two advantages: small size and long lifetime, which makes the LC sensor superior in some applications such as implantable sensors and wearable devices [9].

Previously reported flexible LC pressure sensors offered capabilities for the measurement of physical signals inside and outside the human body. Some LC sensors were fabricated by lithography, etching processes, or screen printing on the polyimide substrate. Bao et al. presented an ultra-small passive sensor whose pressure sensitivity was −1.428 MHz/mmHg for health monitoring and critical care [16]. The sensors were fabricated using a low-cost method of wax printing which etched copper layer coating on top of a flexible polyimide film or a standard lithography method where the copper of 900 nm was thermally evaporated onto a polyimide-coated silicon wafer and then was etched. Moreover, a micro-structured styrene-butadiene-styrene (SBS) elastomer was made by using soft lithography as the pressure-sensitive dielectric layer, but this method needed to replicate the mold pattern repeatedly. Rogers et al. also presented a wireless epidermal sensor of hydration and strain, in which serpentine coil antennas and electrodes were made of photolithography patterned layers of copper deposited by electron beam evaporation on the PI film [1]. Kou et al. showed a wireless wide-range pressure sensor based on a graphene/PDMS sponge for tactile monitoring, which had a pressure sensitivity of −2.2 MHz/kPa [17]. The wireless pressure sensor used a GR/PDMS sponge as the dielectric layer sandwiched by folding the PI substrate with patterned Cu as the antenna and electrode. Huang et al. came up with a sensitive LC wireless pressure sensor with micro-structured PDMS dielectric layers for wound monitoring, in which planar spiral inductors were patterned using conductive ink on a 50 μm thick polyimide by a screen-printing method [18]. The PDMS dielectric layer with inverted pyramid microstructure was obtained by pouring the PDMS mixture into the silicon wafer mold etched through an anisotropic etching process and curing it. The pressure sensitivity of the sensor was −284.9 kHz/mmHg. Nie et al. fabricated a textile-based wireless pressure sensor for human-interactive sensing. It was made of a layer of soft fabric spacers sandwiched between the LC antenna and a ferrite film and had a pressure sensitivity of −0.19 MHz/kPa [19]. The capacitor and inductor were formed on the two sides of the double-sided copper-clad polyimide film using the standard screen-printing method and a wet etching process. Then the two copper layers were connected via through-hole copper plating. In the assembly of the sensor, the antenna and the ferrite film were pasted on both sides of the fabric spacer by applying a layer of double-sided adhesive tape.

LC sensor can be prepared on a PET substrate through the electroplating process. Huang et al. proposed a symmetric LC circuit configuration for passive wireless multifunctional sensors. The sensor circuits on the top and bottom surface of the PET substrate were fabricated through an electroplating process, and the PDMS-based pressure-sensitive membrane with arrays of the inverted micro-pyramid was fabricated using a silicon mold. The sensitivity of the pressure sensor was −48.5 kHz/mmHg [20].

Solution-free vacuum deposition of metals on a PDMS substrate is also a method of preparing LC sensors. Park et al. fabricated a wireless pressure-sensing platform used for parallel signal processing. The metal-based (Al or Au) 2D inductance coil was patterned in a solution-free manner using vacuum deposition, and the lateral electrode, made up of either Al or Ag nanowires, were patterned on the glass or PDMS. The pressure-sensitive element was a PDMS substrate with a pyramid structure [21].

In addition, many groups have contributed to the advancement of implantable flexible LC pressure sensors for the measurement of internal physiological signals of the human body over the past decade.

Piezo inkjet is an effective method for preparing LC inductor coils on a paper substrate. Tse Nga Ng et al. proposed a wireless pressure sensor with a pressure sensitivity of 0.011 MHz/mmHg prepared by inkjet printing to capture the heart pulse waveform. The inductor coil and the capacitor electrodes were patterned using a piezoelectric inkjet to print nanoparticle silver ink on a paper substrate. The porous dielectric layer was fabricated through polymethyl methacrylate microspheres mixed with PDMS and hexane in the specific ratio by weight. A piece of the Ag electrode with PDMS porous dielectric was bonded to the paper substrate with the complementary capacitor electrode and the inductor coil by using a diluted PDMS solution as the glue [22].

Some people used the method of laser cutting metal layers to fabricate LC coils and electrodes. Rogers et al. presented a bioabsorbable wireless pressure sensor with pressure sensitivity of 200 kHz/mmHg for real-time monitoring of human internal pressure [23]. Laser cutting foils of Mg yielded spiral coils, spacers, and the bottom electrode. Adhesion between the Mg spacer and the bottom electrode relied on candelilla wax. The flexible top Zn electrode was laminated on the Mg spacer.

Electron beam evaporation of metal particles was also commonly used in the preparation of LC coil antennas. Aleksi Palmroth et al. proposed a wireless bioabsorbable pressure sensor embedded in orthopedic implants with a pressure sensitivity of −6 kHz/mmHg used to monitor the healing process and detect complications. Conductor patterns were fabricated by the electron-beam evaporating the magnesium pellets onto the poly (DTE carbonate) substrate through 3D printed masks, and the compression-molded poly (DTE carbonate) spacer was used between two substrates to form a sandwich structure [24].

In the above-mentioned work, most flexible LC pressure sensors were prepared on flexible polymer films such as PI or paper. Most flexible polymer films don’t have excellent stretchability due to their high Young’s modulus, which makes it difficult for them to conformally attach to arbitrarily curved and moving surfaces (such as joints) to continuously monitor bending. On the other hand, it is difficult to compress the dielectrics of most sensors such as PDMS because of their high Young’s modulus, resulting in the low-pressure sensitivity of the sensor. In addition, the above-mentioned work relied on the metal coil connected to the vector network analyzer to read the information of the LC sensor. Because the VNA is bulky and inconvenient to move, it is greatly restricted in daily applications.

In order to solve the above-mentioned problems, we fabricated a flexible LC pressure sensor with Ecoflex (0030, Smooth-on Inc., Macungie, PA, USA) as the substrate that can be conformally attached to the arbitrarily curved and moving surface of the skin to monitor bending long-term. At the same time, we used a method of metal mold replication to prepare Ecoflex microcolumns as the dielectric of the pressure sensor. Because the gap in the dielectric constant value between Ecoflex material and PDMS material is small, and the Young’s modulus of the Ecoflex polymer is smaller than that PDMS polymer, the LC pressure sensor with Ecoflex microcolumns as the dielectric has higher pressure sensitivity than most of the previously reported work. The antenna can be utilized to transmit energy and information, and it has been widely utilized in RFID technology as one of the main components of the portable readout device. Here, we designed a coplanar waveguide-fed monopole antenna used to read the information of the LC sensor, whose bandwidth below −10 dB is wide enough to cover the probable range of frequency variation of the sensor.

In this paper, we proposed a flexible LC pressure sensor with high-pressure sensitivity for the measurement of the physical activity signals. Firstly, the finite element analysis method helped to design and optimize the flexible LC pressure sensor to obtain high sensitivity. Secondly, inspired by the traditional Chinese dough figurines, the embossing method was proposed to prepare the required LC coil. At the same time, we used a simple metal mold replication method to prepare the dielectric of the LC pressure sensor. In order to obtain high-pressure sensitivity, we choose Ecoflex material with a lower Young’s modulus to replace the commonly used PDMS material. We proposed a transfer-paste method to prepare the flexible LC pressure sensors with a sandwich structure based on different flexible substrates, which can meet the needs of a variety of different bonding surfaces, and they can be integrated as flexible LC sensing arrays to solve the problem of complex wired sensor arrays. Thirdly, in order to be easily integrated with the radio frequency signal processing circuit to meet the development direction of the miniaturization and integration of the LC reader system, we designed a coplanar waveguide-fed monopole antenna used to read the information of the LC sensor, expanding the types of readout devices of sensors, and improving the vertical readout distance and horizontal readout range. Finally, to confirm the practicality of the fabricated pressure sensor, we used medical tape to stick it on different parts of the human body to measure physical activity, such as limb bending and facial muscle movement.

The pressure sensor with high sensitivity in this paper can be used to monitor the distribution of pressures, bending of physical joints, and swelling in the injured area. The planar monopole antenna can be integrated with RF signal processing circuits as a miniaturized portable readout device. The whole-body wireless pressure sensing network will be constructed by sticking the pressure sensor to various parts of the human body and integrating the readout device on the clothes, which has great application potential in the monitoring and analysis of athletes’ physical motion signals and medical care.

## 2. Design, Fabrication, and Readout Principle of the LC Sensor

### 2.1. Design of the LC Sensor

In this work, we introduced a flexible wireless passive LC pressure sensor that was attached to different body sites for the measurement of human body motions, as illustrated in Figure 1a. When the state of limb motion changes, the resonance frequency of the flexible LC pressure sensor changes. A readout device connected to the vector network analyzer is used to wirelessly interrogate the resonance frequency of the sensor. The vector network analyzer can transmit a series of sweep signals which contain the resonance frequency of the sensor to the readout device, which is magnetically coupled with the sensor. When the frequency of the electromagnetic signal in the readout device corresponds to the resonance frequency of the sensor, electromagnetic energy was maximally absorbed by the sensor. Therefore, the input return loss ($S_{11}$) of the readout device at this resonance frequency reaches the minimum [18]. By analyzing the resonance frequency shift of the sensor, we can judge the different motion states of the body.

The flexible LC pressure sensor is created by stacking a micro-column dielectric between the two capacitor plates in a sandwich structure [17], as depicted in Figure 1b. Under applied pressure, the separation distance between the capacitor plates is reduced, so the effective capacitance is increased, and the resonance frequency of the sensor shifts down.

To evaluate the sensitivity and optimize the design of the flexible LC pressure sensor, the finite element simulation of the sensor under different pressures was carried out with the COMSOL software (Multiphysics 5.4, COMSOL Inc., Stockholm, Sweden). In order to verify the frequency sweep range in simulation, theoretical formulas were utilized to calculate the inductance and capacitance of the LC pressure sensor to obtain its resonance frequency. The size parameters of the coil antenna and the micro-column dielectric are shown in Figure 1c,d, respectively.

For square planar inductors, the approximate formula of the current sheet derived from the basic electromagnetic theory is as follows [25]:(1)$$L=\left[\frac{u_{0}N^{2}\left(\frac{d_{in}+d_{out}}{2}\right)c_{1}}{2}\right]\left[ln(\frac{c_{2}}{\alpha }\right)+c_{3}\alpha +c_{4}\alpha^{2}]$$
where $c_{1}=1.27,\;c_{2}=2.07,\;c_{3}=0.18,$ and $c_{4}=0.13$ are constants based on the geometrical layout of the square spiral; α is the fill ratio defined by $\alpha =\left(d_{out}-d_{in}\right)/\left(d_{out}+d_{in}\right)$; $u_{0}$ is the permeability of the vacuum; and N is coil turns. When the size parameters of the coil antenna are put into the formula of the inductance, the value of the inductance is approximately $1.829\times 10^{-7}\;$H.

As shown in Figure 1b, the LC sensor can be equivalent to an RLC series circuit consisting of an inductor, a capacitor, and a resistor. Here, the U-shaped conductor is equivalent to the connecting wire between the inductor and the capacitor. Therefore, the influence of the U-shaped conductor connection is ignored in the calculation of the equivalent capacitance. In the presence of a micro-column dielectric, the effective capacitance ($C_{\mathrm{eff}}$) is computed with a modified parallel plate capacitor as shown in the formula below [16]:(2)$$C_{eff}=\left(\frac{V_{air}}{V_{air}+V_{elastomer}}\varepsilon_{0}+\frac{V_{elastomer}}{V_{air}+V_{elastomer}}\varepsilon_{elastomer}\right)\frac{A_{eff}}{d_{sep}}$$
in which $\varepsilon_{0}$ and $\varepsilon_{elastomer}$ are dielectric constants of the vacuum and elastomer, respectively. The weighted ratio for them is based on the volume proportion of air $V_{\mathrm{air}}$ and elastomer $V_{elastomer}$ between the capacitor plates. $A_{eff}$ and $d_{sep}$ are respectively the relative area and the separation distance between capacitor plates. When the size parameters of the Ecoflex micro-column dielectric are put into the formula of the capacitance, the value of the capacitance is approximately $6.702\times 10^{-13\;}$F.

The resonance frequency of the flexible LC pressure sensor can be expressed as [9]:(3)$$f_{S}=\frac{1}{2\pi \sqrt{LC}}$$
where *L* and *C* are the inductance and capacitance of the sensor tank, respectively. When the above-calculated values of inductance and capacitance are brought into the calculation expression of the resonant frequency, the calculated resonance frequency of the sensor is approximately 455 MHz.

The sensitivity of the flexible LC pressure sensor was studied with the finite element simulation using the Electromagnetic Waves module and Solid Mechanics module of the COMSOL software. The Solid Mechanics module was taken advantage of simulating the compression of the sensor caused by external pressure, and the Electromagnetic Waves module was used to simulate the response of the readout antenna in the readout system. The meshed model diagram of the readout system of the sensor in the finite element simulation is shown in Figure 2a.

We used a finite element analysis to optimize the size parameters of the coil antenna and microcolumn dielectric of the LC sensor. As shown in Appendix A, the simulation results show that the pressure sensitivity of the LC sensor increases as the area of the capacitor plate increases. At the same time, the smaller the side length of the micro-column is, the higher the pressure sensitivity of the LC sensor is. Due to the limitations of practical applications and manufacturing processes, the size of the coil antenna and the microcolumn are limited to the centimeter and millimeter levels, respectively. The optimization of each parameter requires the computer to operate about 12 h, and the resulting program occupies about 30–40 G of computer memory.

The materials of the substrate, inductor coil, and micro-column were respectively set as polyimide (PI), copper, and polydimethylsiloxane (PDMS), as illustrated in Figure 2b. In the simulation, the resonant frequency of the flexible LC PDMS pressure sensor is found to decrease with the increase of the applied pressure, as depicted in Figure 2c. From Figure 2d, it is clear that the measured resonant frequency is a linear function of pressure over a pressure range of 0–5 kPa with a sensitivity of −0.76 MHz/kPa. In order to improve the sensitivity and wearing comfort of the sensor, we used silicone elastomer (Ecoflex) instead of PDMS and PI due to its low Young’s modulus, which makes it easier to be compressed [26,27]. The PDMS and Ecoflex material values are given in Appendix A, respectively. Figure 2e shows the simulation model of the flexible LC pressure sensor based on Ecoflex material, and its resonance frequency curves under different pressures are depicted in Figure 2f. Its resonant frequency changes according to a linear relationship of applied pressure in the range of 0–2 kPa with the sensitivity of −1.5 MHz/kPa. The pressure sensor with Ecoflex micro-column as a capacitor dielectric shows the improved sensitivity compared with that using PDMS as a micro-column dielectric, which is in good agreement with the above analysis. The resonance frequency of the sensor is 455 MHz calculated by the theoretical formula and 404 MHz obtained from the finite element simulation. The above values are fairly close and differ little, which indicates that the finite element simulation has high accuracy in the performance assessment of the LC pressure sensor.

The stress distribution of the sensor with an Ecoflex micro-column as the dielectric, under an applied pressure of 2000 Pa over the surface, is shown in Figure 2g. The stress is evenly distributed on the lower surface of the upper substrate, which indicates the pressure sensor has high structural stability.

### 2.2. Fabrication of the LC Sensor

A method of metal template replication was used to prepare the Ecoflex micro-column dielectric, which was cheaper than the preparation process of the microstructure dielectric and simpler than the fabrication method of the composite dielectric; the manufacturing process is shown in Figure 3a. Firstly, we stuck the 100 μm thick polyimide film on the glass plate with double-sided adhesive. The aluminum template (Hangyi Hardware, CN) was soaked in the polyvinyl alcohol solution (PVA, 50 mg/mL, Shandong Yousuo Chemical Technology Co., Ltd., Shandong, China) and taken out by tweezers. We used a blower to remove the redundant PVA solution on the surface of the template and put it on the PI film. Then, the template was dried in the oven (DHG-9030A, Shanghai Jingqi Instrument and Device Co., Ltd., Shanghai, China) at 45 °C for 5 min. Secondly, we prepared the Ecoflex mixture, in which the ratio of EcoflexA glue: EcoflexB glue is 1:1. The Ecoflex mixture was placed in a vacuum chamber (Linhai Tanshi Vacuum Device Co., Ltd., Linhai City, China) for 20 min to remove bubbles. Thirdly, the Ecoflex mixture was spin-coated (100 rad/s, 30 s) in the template and cured at 80 °C for 20 min. Finally, we peeled off the Ecoflex micro-columns from the template and used a paper cutter to cut them to the size needed.

Inspired by the embossing technology of the Chinese traditional dough sculpture process, we proposed an embossing process to prepare the LC coil antenna, which was simple and low-cost compared to the reported works. Figure 3b illustrates the production process of the LC coil antenna, and the physical map is shown in Appendix A. First, we stack the lower laminate, copper conductive tape (100 μm, Site Jie-Tech Co., Ltd., Shanghai, China), coil antenna template (Duofen Craft Product Co., Ltd., Shanghai, China), and upper laminate in order. Second, we put it in the embossing device (Duofen Craft Product Co., Ltd., China) and turned the handle of the device. At last, the LC coil antenna was peeled off from the metal template using a tweezer.

In this paper, we presented a transfer-paste method for the fabrication of the flexible LC pressure sensor, which can achieve sensors based on various dielectrics and substrates. It provided a new idea for the preparation of LC sensors. The manufacturing process of the sensor is depicted in Figure 3c. First, the copper LC coil was transferred between the two Ecoflex substrates and bonded with silicone glue. Next, the Ecoflex micro-column dielectric was pasted between the capacitor plates to form the pressure sensor with a sandwich structure. Finally, the Ecoflex substrate can be replaced with a PI or PDMS substrate, and the Ecoflex micro-column can be substituted using the PDMS micro-column. Various flexible LC pressure sensors are displayed in Figure 3d–f, respectively.

### 2.3. Performance Test and the Readout Principle of the Copper Ring Device

To wirelessly interrogate the flexible LC pressure sensor, a readout copper ring (diameter: 3 cm, line width: 3.5 mm) connected to the vector network analyzer was near-field electromagnetic coupled with the sensor, as shown in Figure 4a, and the equivalent circuit diagram is depicted in Figure 4b. The resonant frequency of the sensor was remotely monitored through the minimum of the input return loss ($S_{11}$) in the readout device using the vector network analyzer.

The impedance of the readout copper ring ($Z_{R}$) measured by the vector network analyzer can be expressed as [1]:(4)$$Z_{R}=R_{R}+j\omega L_{R}+\frac{\omega^{2}M^{2}}{Z_{S}}$$

The impedances of the flexible LC pressure sensor ($Z_{S}$), the resistance ($R_{R}$), and inductance ($L_{R}$) of the readout copper ring all influence $Z_{R}$.

The impedance of the sensor is [1]:(5)$$Z_{S}=R_{S}+j\omega L_{S}-j\frac{1}{\omega C_{s}}$$

The coupling coefficient (*k*) and the quality factor (*Q*) of the sensor can be expressed as [1]:(6)$$k=\frac{M}{\sqrt{L_{R}L_{S}}}$$
(7)$$Q=R_{S}\sqrt{\frac{L_{S}}{C_{S}}}$$
where *M* is the mutual inductance between the readout coil and the pressure sensor.

By using the Equations (4)–(7) the impedance of the readout coil can be further expressed as [28]:(8)$$Z_{R}=R_{R}+j2\pi f_{R}L_{R}\left[1+\frac{k^{2}{\left(\frac{f_{R}}{f_{S}}\right)}^{2}}{1+j\frac{1}{Q}\frac{f_{R}}{f_{S}}-{\left(\frac{f_{R}}{f_{S}}\right)}^{2}}\right]$$
where $f_{R}$ and $f_{S}$ are the intrinsic resonance frequency of the readout coil and the LC pressure sensor, respectively.

For a one-port measurement system, the input return loss, $S_{11}$ parameter can be expressed as [9]:(9)$$S_{11}=\frac{Z_{R}-Z_{0}}{Z_{R}+Z_{0}}{|}_{Z_{0}=50\Omega }$$

According to Equations (8) and (9), the measurement depends critically on the coupling coefficient between the pressure sensor and the readout copper ring. Therefore, the relative position between them is important [1].

To examine the influence of different vertical readout distances, the readout copper ring was placed underneath the LC pressure sensor, and the sensor was placed on the acrylic tables with different heights. Changes in the resonant frequency of the sensor are shown in Figure 4c. As the vertical reading distance increases, the signal strength of the resonance frequency that was read by the copper ring gradually decreases. Its resonance frequency remains at 350 MHz, which is fairly close to the results of the theoretical calculation (455 MHz) and the finite element simulation (404 MHz). This not only indicates that the readout copper ring has stable readability within the vertical readout distance of 0–2.5 cm but also further shows the accuracy of the finite element simulation used to evaluate sensor performance.

The effect of different horizontal displacements is shown in Figure 4d. The pressure sensor was placed on the acrylic table, and the readout copper ring was placed underneath the sensor at a fixed distance of 5 mm. The change in resonance frequency is observed for a horizontal displacement of up to 1.5 cm and is followed without a resonance frequency peak beyond this limit.

The rotation of the readout copper ring was adjusted using acrylic tables with different angles. Figure 4c shows that the measured resonance frequencies remain at 340 MHz for rotations between 0 to 90°, and the signal amplitude of them gradually decreases.

These measurements demonstrate a robust reading range of vertical distances of 0–2.5 cm, horizontal displacements of 0–1.5 cm, and rotation angles of 0–90°, in which reliable wireless detection can be achieved, with little shift in the resonance frequency.

## 3. Results and Discussion

### 3.1. Pressure Sensitivity Test of the Flexible LC Pressure Sensor

In order to test the sensitivity of flexible LC pressure sensors, the metal weights with different masses were used to apply pressures to the sensors, and a small plastic dish was placed between the metal weight and the sensor to avoid producing additional parasitic capacitance, as depicted in Figure 5a. Under different applied pressures, resonance frequency curves of sensors based on Ecoflex, PI, and PDMS are respectively illustrated in Figure 5b–d.

The resonance frequency of the flexible LC pressure sensor based on Ecoflex substrate and micropillar dielectric is 360.6 MHz in the case of only the small plastic dish applied. With the applied pressure increasing, its resonance frequency decreases linearly, reaching 311 MHz under the pressure of 23,760 Pa. The LC pressure sensor has the PI substrate and Ecoflex microcolumn dielectric, whose resonant frequency reduces from 384 MHz to 323 MHz when the pressure increases from 675 Pa to 23,760 Pa. For the LC pressure sensor of the PDMS substrate and dielectric, its resonant frequency decreases linearly from 385 MHz at a pressure of 675 Pa to 368 MHz at a pressure of 23,760 Pa.

Due to the different dielectric constants existing in various flexible substrates, the initial values of the resonance frequency among the sensors are different. Meanwhile, considering the effect of the dielectric properties of the small plastic dish on the sensor, the initial resonance frequency of the LC sensor is placed in the presence of the small plastic dish to only represent the effect of different pressures on the resonance frequency of the sensor.

Figure 5e shows that the measured resonance frequencies of various sensors have a linear relationship with applied external pressures in the range of 0–23,760 Pa. It is clear that the sensor with a PI substrate and Ecoflex micro-column dielectric has a sensitivity of −2.4 MHz/kPa, and when the substrate material is changed into Ecoflex, the sensitivity is −2.2 MHz/kPa, which is roughly the same as shown in Figure 5e. In contrast, the sensor based on PDMS has a low sensitivity of -0.6 MHz/kPa. The values of the sensitivity of the pressure sensors based on different materials obtained through experiments and simulations are shown in Table 1. Through the data comparison in Table 1, it is further verified that the finite element simulation designed in this paper can be used to accurately evaluate the sensor performance. Based on Ecoflex, the sensitivity of the sensor obtained through the experiment is higher than that achieved by simulation, mainly due to the difference between the material parameters in the simulation and the actual situation.

The flexible LC pressure sensor designed based on Ecoflex material in this paper not only has higher pressure sensitivity than most sensors reported in the existing works (as shown in Table 2) but also fits human skin well and has higher wearing comfort compared to the sensors based on PI substrate and PDMS substrate [20,29].

### 3.2. Flexible LC Pressure Sensing Array

The flexible LC pressure sensors with different resonance frequencies were integrated into a pressure sensing array, as shown in Figure 6a. Each unit in the sensing array is designed with a specific resonant frequency from which it can be individually addressed without any crosstalk [16].

In order to solve the problem of complicated wiring of the traditional wired array, we have fabricated a sensing array by implementing three types of sensors for the sensing units; it can be detected in Figure 6b at a distance of 5 mm from a readout copper ring (diameter: 4 cm, Line width: 5 mm) in the air environment. When we apply pressure to sensing unit 1, its resonant frequency decreases from 340 MHz to 323 MHz as the pressure increases from 0 Pa to 23,760 Pa, and the frequencies of the sensing unit 2 and the sensing unit 3 remains at 284.5 MHz and 269.6 MHz, respectively. The results show that the frequency curves of all three sensors in the sensing array can be simultaneously detected, and the resonance frequency of each sensor varies individually as the pressure increases without interfering with each other.

To prove that the transfer-paste method can be used to prepare pressure sensor arrays with excellent performance, we also fabricated a sensing array with two types of antennas, as illustrated in Figure 6c,d. When pressure is applied to one of the sensing units, its resonance frequency decreases with increasing pressure, and the frequency of the other sensing unit remains constant. The resonant frequency of sensing unit 2 in the array is about 360 MHz at 0 Pa, which is consistent with its intrinsic resonant frequency when used as a single sensor. This further demonstrates that the electromagnetic crosstalk between units of the sensing array is negligible, so each unit can work independently.

### 3.3. Performance Tests of the Monopole Antenna for Reading the Flexible LC Pressure Sensor Information

In this paper, we designed a coplanar waveguide-fed monopole antenna used to read the information of the LC sensor, whose bandwidth below −10 dB was wide enough to cover the probable range of frequency variation of the sensor [30,31,32,33,34].

Usually, when the frequency of the planar monopole antenna is several hundred MHz, the length of it can reach tens of centimeters, which does not meet the demand of the portable readout devices for miniaturized antennas. At the same time, it must cause the scattered distribution of the magnetic field, so the information of the LC sensor will be difficult to be read successfully.

In order to reduce the size of the monopole antenna, we designed a planar monopole antenna with a spiral structure. Its size parameters are shown in Appendix A.

The monopole antenna is shown in Figure 7a. Its intrinsic resonance frequency is 287 MHz obtained by HFSS simulation, as illustrated in Figure 7b. To verify the ability of the monopole antenna to read the information of the LC sensor, the readout performance of the monopole antenna was tested when it was placed in different vertical distances, horizontal displacements, and rotation angles [1,34].

Figure 7c shows that the resonant frequency curves of the sensor were obtained by the monopole antenna located at different vertical distances. Its intrinsic resonance frequency maintains 270 MHz, and the frequencies of the sensor remain at 350 MHz, which is almost the same as the frequency of the sensor accessed by the readout copper ring. This shows that the planar monopole antenna can accurately read the information of the LC sensor. At the vertical distance of 4 cm, the monopole antenna can still weakly read the signal from the sensor, which is higher than the 2.5 cm maximum readout distance of the copper ring.

The formula for calculating the wavelength of the planar monopole antenna is
(10)$$\lambda_{m}=\frac{c}{f_{m}}$$
where $c=3\times 10^{8}\;m/s$ is the speed of light and $f_{m}=287\;\mathrm{MHz}$ obtained from the HFSS simulation is the intrinsic resonance frequency of the monopole antenna. The wavelength corresponding to the intrinsic resonance frequency is 1 m, calculated with Formula (10).

The formula of the distance of the antenna’s inductive near-field zone is
(11)$$R=0.62\sqrt{\frac{L^{3}}{\lambda_{m}}}$$
where $L=\lambda_{m}/4$ is length of the monopole antenna. The distance of the inductive near-field zone (*R*) is 8 cm, calculated from Formula (11).

As shown in Figure 7c, the maximum readout distance of the planar monopole antenna is 4 cm, so it reads the information of the LC sensor by inductive coupling within this range. Its surface current intensity and magnetic field intensity are shown in Appendix A, respectively.

From Appendix A, we can see that the current is mainly distributed in the outer ring of the monopole antenna with the spiral structure. The magnetic field is concentrated around the monopole antenna.

Compared to the copper ring, the monopole antenna can read the information of the LC sensor reaching a distance of 4 cm. The reason is that the magnetic induction generated by the planar monopole antenna is higher than the copper ring. According to Biot-Savart law, the strength of the magnetic induction at a certain point is equal to the sum of the magnetic induction strength of each current element, expressed as
(12)$$B=\frac{\mu_{0}}{4\pi }\int_{A}^{C}\frac{Idz\sin\theta }{r^{2}}$$
where *B* is magnetic induction, $\mu_{0}$ is vacuum permeability, Idz is the current element, *r* is the distance from the current element to the point, and θ is the angle between the current element and the position vector of the current element to the point. The spiral structure of the monopole antenna can be equivalent to multi-turn coils, which have a larger effective integration range than a single-turn copper ring.

To examine the influence of horizontal displacement, the monopole antenna was placed underneath the sensor at a fixed distance of 1 mm, and the measured resonance frequency curves were depicted in Figure 7d. The obtained resonance frequency of the sensor remains at 350 MHz in the horizontal displacements range of 0–8 cm, which is larger than the range of 0–1.5 cm copper ring.

It is demonstrated in Figure 7e that the rotation of the monopole antenna is adjusted using acrylic boards with different bending angles. The measured frequency remains at 350 MHz for rotations ranging from 0 to 90°, and the signal strength gradually decreases with the increase of the rotation angle. Compared to the readout angle range of the 30° in reported research [34], the monopole antenna in this paper exhibits a larger range of readout angles, which can weakly read out the resonance frequency when the rotation angle reaches 90°.

There is a robust range of the vertical distance from 0 to 4 cm, the horizontal displacement from 0 to 8 cm, and the rotation angle from 0 to 90° for the monopole antenna, where it can read the information of the sensor accurately, as shown in Table 3.

In order to further verify that the monopole antenna can accurately read the information of the LC sensor, it was used to measure the sensitivity of the pressure sensor, as shown in Figure 7f. When the pressure increases from 0 Pa to 23,760 Pa, the resonant frequency of the LC sensor decreases from 342 MHz to 279 MHz in a nearly linear trend, as illustrated in Figure 7g. Compared with the tested results of the copper ring, the resonance frequency of the sensor measured by the monopole antenna is lower, but the sensitivity values are basically the same. The main reason for this difference in resonant frequency is that the proximity between the sensor and the monopole antenna creates additional parasitic capacitance.

Figure 7h shows that the monopole antenna is used to read the information of the LC pressure sensing array. When pressure is applied to sensing unit 2, its resonance frequency decreases from 340 MHz to 320 MHz, and the frequencies of sensing unit 1 and the monopole antenna remain constant. This shows that the monopole antenna can be excellently utilized to read the information of the LC pressure sensing array.

By comparing the information of the sensors read by two readout devices, the measured results of using a planar monopole antenna as the readout device basically agree with the test results of the copper ring, which further verifies the feasibility of the application of the monopole antenna in the readout system of the wireless passive LC sensor. The feasibility of the coplanar waveguide fed monopole antenna as a reader antenna has been verified in GHz frequency band applications [34]. In this paper, we verified the feasibility of the monopole antenna used to read the information of the LC sensor in the hundred MHz range.

### 3.4. Flexible LC Pressure Sensor for Monitoring Physical Motion Signals

With life expectancy and various health risks rising simultaneously, flexible wireless wearable electronic devices have been studied extensively and used for continuous health monitoring and instant diagnoses to improve people’s health [18]. Currently, people perform various physical activities to exercise their bodies. In order to ensure a suitable extent of motion and avoid injury, it is very important to monitor the body’s limb motion. The proposed flexible LC pressure sensor, based on Ecoflex, in this paper, has high sensitivity and conformally contracts with human skin, so it is suitable for the measurement of various limb motions. The tests are based on the principle that the pressure produced by physical activity changes the resonance frequency of the flexible LC pressure sensor.

The sensor was attached to a finger to measure the finger bending movements [17], as illustrated in Figure 8a. Each finger bending state corresponds to a resonance frequency curve. When the finger was bent from the horizontal state (0°) to the vertical state (90°), the resonant frequency decreased from 322.8 MHz to 225 MHz. The resonance frequency measured by the monopole antenna was changed from 374.4 MHz to 279.1 MHz, as shown in Figure 8b. The relationship curves between the resonant frequency and the bending angle are shown in Figure 8c, where the frequency measured by the copper ring is lower than the monopole antenna, but the variation tendencies are the same.

This work suggests that the planar monopole antenna acting as a reader antenna of the LC sensor can be used to monitor human activity signals, although it is influenced by the dielectric properties of the human skin to a certain extent. This also provides an optional readout antenna for a miniaturized, integrated mobile terminal readout device in an LC sensing system, which may be used in medical monitoring systems.

To further investigate the performance of the sensor in terms of human movement monitoring, we attached the sensor to the face (center of the forehead) to monitor the facial muscle movements of a smile or frown, as depicted in Figure 8d–g. At the same time, the sensor can well monitor the bending movement of the wrist, as shown in Figure 8h,i.

People inevitably have bumps during exercise and daily life, resulting in swelling of the skin. The degree of swelling of the skin should be monitored at all times to help with time-consuming and effective treatments of the swollen area. We used the medical tape to stick the flexible LC pressure sensor on the surface of the balloon and use the balloon’s expansion to simulate the swelling of the skin, as illustrated in Figure 8i. As the inflation degrees of the balloon increase, the resonant frequency of the sensor gradually decreases.

Therefore, the sensor can wirelessly detect limb bending and facial muscle movements, which gives it the potential to construct a body wireless sensor network for motion monitoring and pressure mapping.

## 4. Conclusions

In summary, we proposed a sandwich-structured flexible LC pressure sensor based on an Ecoflex micro-column as a dielectric, which is sandwiched between the two capacitor plates of the coil antenna. To guide the design and optimization of the sensor, a finite element simulation is used to evaluate the performance of the sensor. We presented a simple embossing process to emboss copper conductive tape to prepare a low-cost coil antenna and a transfer-paste method to fabricate the pressure sensor. The resonance frequency of the sensor changes as a linear function of applied pressure in a range of 0–23,760 Pa with a sensitivity of −2.2 MHz/kPa that is higher than most of the sensors reported in the existing works. We further demonstrated its potential to be integrated into a pressure sensing array with excellent performance. In addition, we also designed a planar monopole antenna to expand the types of readout devices and increase readout distance. The flexible LC pressure sensor can be used to measure various physical activity signals for continuous health monitoring. Looking to the future, the sensor can be combined with an external readout circuit to construct a portable wireless real-time monitoring system. Furthermore, when combined with various chemical materials and biomedical technologies, the LC sensor can be constructed as a multifunctional sensing device, with potentially essential applications in continuous wireless monitoring of multiple physiological parameters for biomedical research and patient care.

## Figures and Tables

**Figure 1 micromachines-12-00976-f001:**
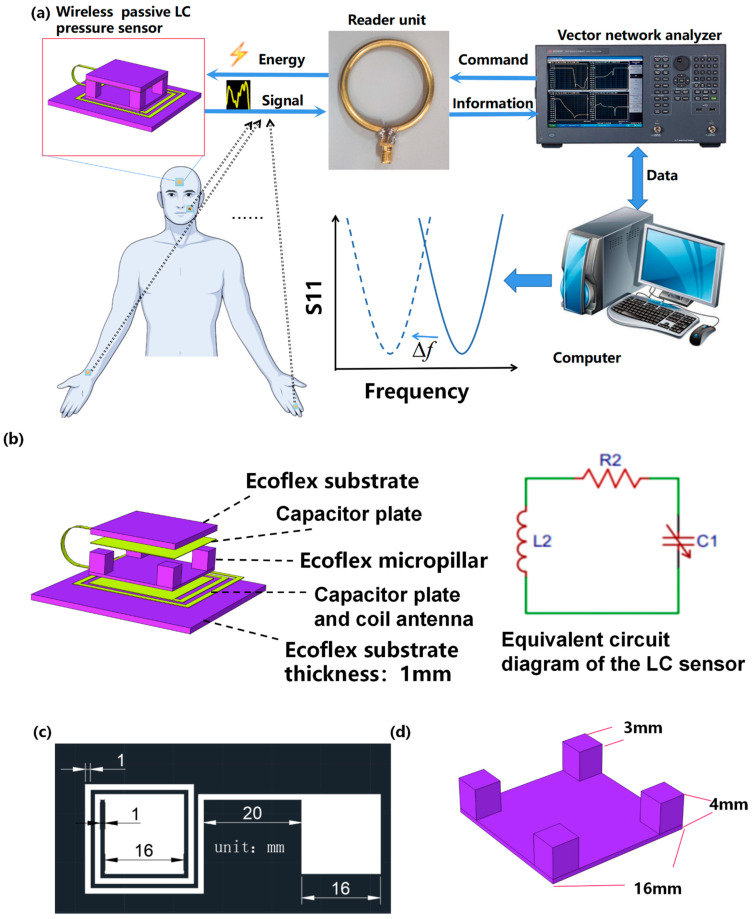
(**a**) Illustration of the flexible LC pressure sensor for the measurement of physical movement signals. (**b**) Exploded view schematic diagram of the flexible LC pressure sensor. (**c**) Schematic diagram of the coil antenna, and (**d**) micro-column dielectric with marked size parameters.

**Figure 2 micromachines-12-00976-f002:**
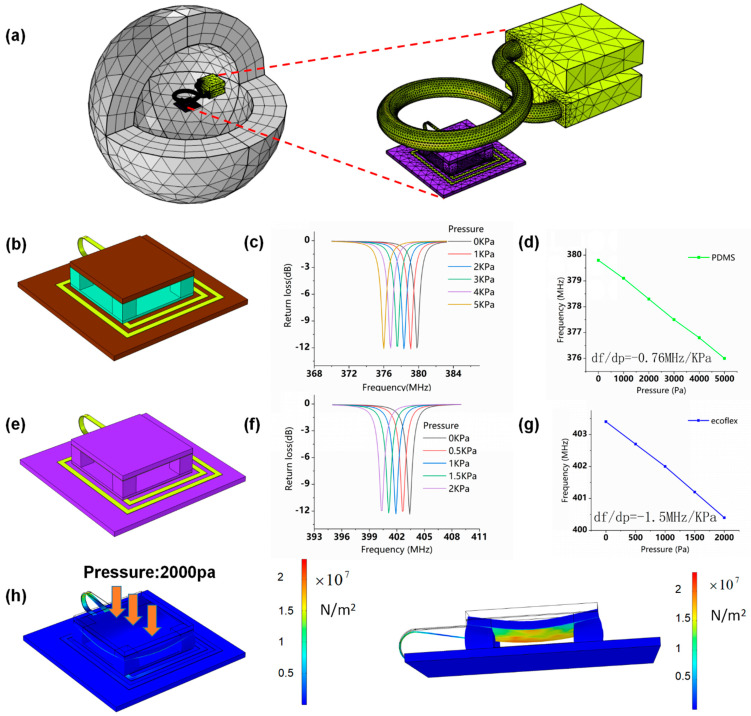
(**a**) The finite element simulation model of the readout system of the flexible LC pressure sensor. (**b**) Simulation model, (**c**) resonance frequency curves, and (**d**) the pressure sensitivity curve of the flexible LC pressure sensor based on PDMS micro-column dielectric and PI substrate. (**e**) Simulation model, (**f**) resonance frequency curves, and (**g**) the pressure sensitivity curve of the flexible LC pressure sensor with Ecoflex micro-column dielectric sandwiched between Ecoflex substrates. (**h**) Simulated stress distribution when the pressure being applied on the upper capacitor plate of the LC sensor based on Ecoflex material is 2000 Pa.

**Figure 3 micromachines-12-00976-f003:**
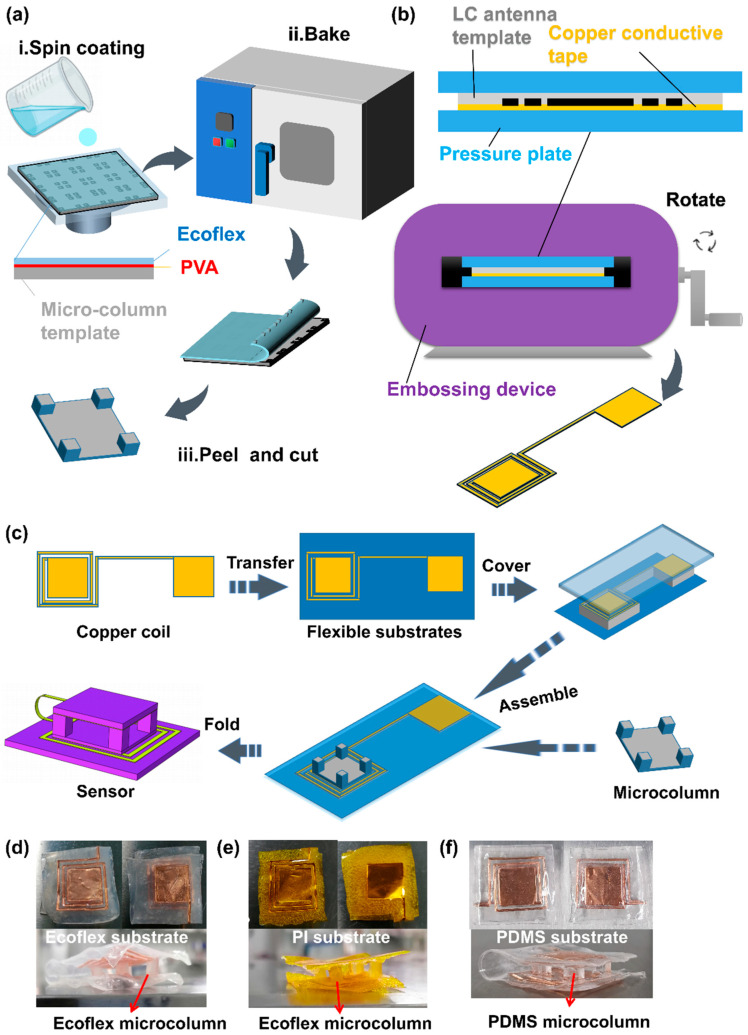
Flow charts of the fabrication of the micro-column dielectric. (**a**), the LC coil antenna (**b**), and the flexible LC pressure sensor (**c**). The physical picture of the sensor with Ecoflex micro-column and substrate (**d**), Ecoflex micro-column and PI substrate (**e**), and PDMS micro-column and substrate (**f**).

**Figure 4 micromachines-12-00976-f004:**
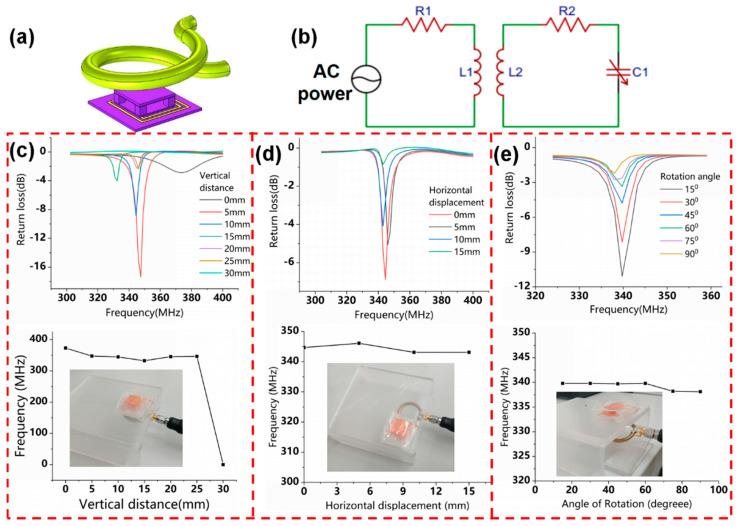
(**a**) Schematic diagram of the readout system of the flexible LC pressure sensor. (**b**) Equivalent circuit diagram of the readout system. Frequency response curves of the flexible LC pressure sensor were measured as the readout copper ring was placed in different vertical distances (**c**), horizontal displacements (**d**), and rotation angles (**e**).

**Figure 5 micromachines-12-00976-f005:**
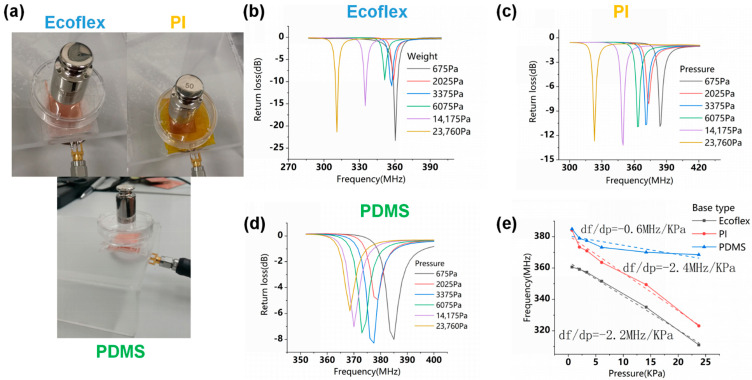
(**a**) Physical images of sensitivity test using various flexible LC pressure sensors. Frequency response curves of the sensor based on Ecoflex micro-column and substrate (**b**), Ecoflex micro-column and PI substrate (**c**), and PDMS micro-column and substrate (**d**), which were applied different pressures. (**e**) The comparison of the pressure sensitivity of various flexible LC pressure sensors.

**Figure 6 micromachines-12-00976-f006:**
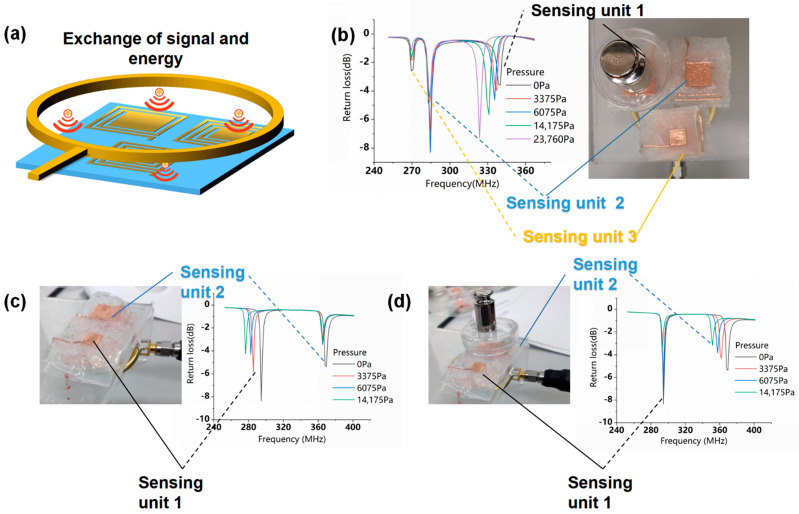
(**a**) Schematic diagram of the readout system of the flexible LC pressure sensing array. (**b**) Frequency response curves of the array with three LC pressure sensors under different pressures applied to the sensing unit 1. Frequency response curves of the array with two LC pressure sensing units under different pressures applied to the sensing unit 1 (**c**) and the sensing unit 2 (**d**).

**Figure 7 micromachines-12-00976-f007:**
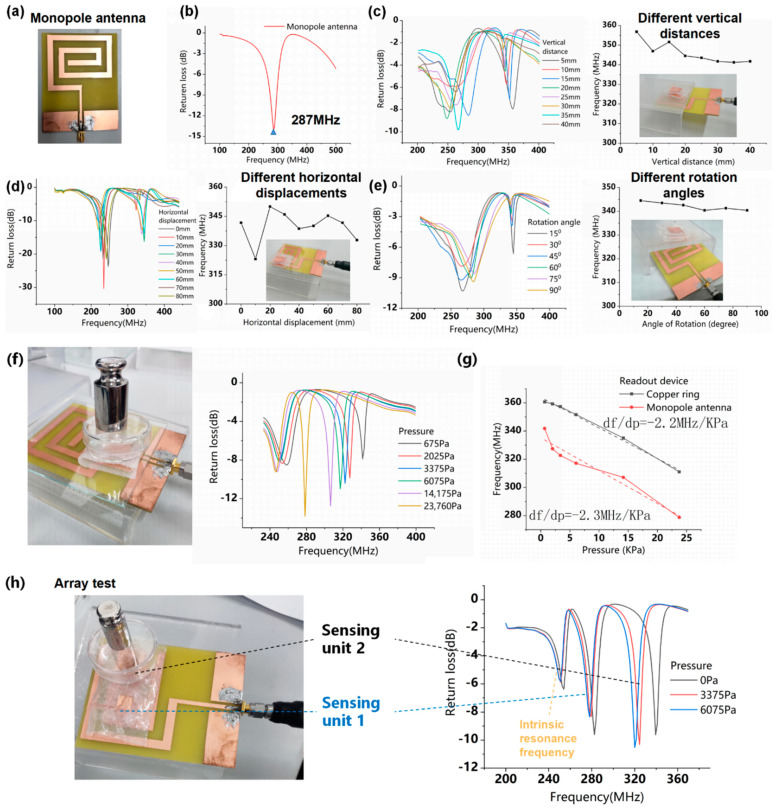
(**a**) Physical drawing of the monopole antenna. (**b**) The intrinsic resonance frequency of the monopole antenna was obtained by the HFSS simulation experiment. When the monopole antenna was located at different vertical distances (**c**), horizontal displacements (**d**), and oriented at different angles (**e**), the resonance frequencies of the sensor were measured. (**f**) The frequency response of the sensor under applied increasing pressure was read out by the monopole antenna. (**g**) Comparison of the sensitivities of different pressure sensors measured by two readout devices. (**h**) Frequency response of the pressure sensing array with sensing unit 2 under applied pressure was measured by the monopole antenna.

**Figure 8 micromachines-12-00976-f008:**
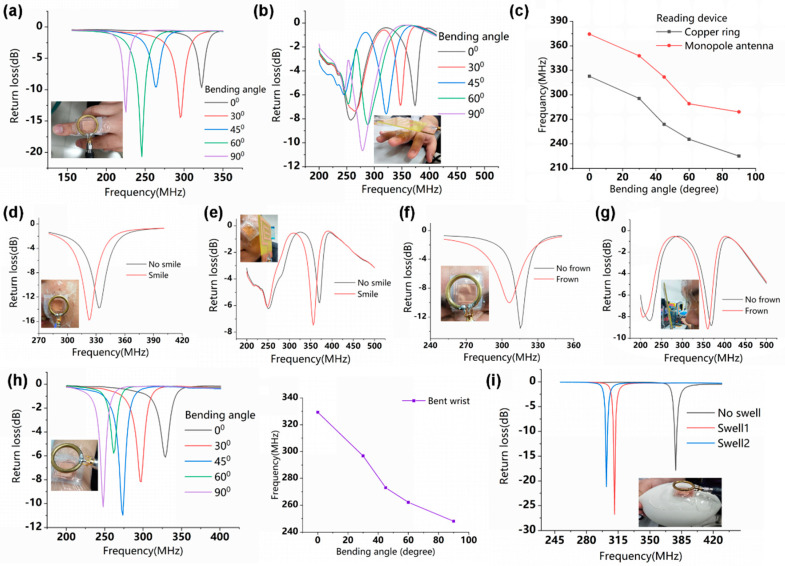
Resonant frequency curves of sensors corresponding to different finger bending states were measured by the readout copper ring (**a**) and the planar monopole antenna (**b**). (**c**) The relationship curves between the resonant frequency of the sensor and different finger bending angles. Resonance frequency curves corresponding to the facial smile (**d**,**e**), frown (**f**,**g**), wrist bending (**h**), and balloon inflation (**i**).

**Table 1 micromachines-12-00976-t001:** Comparison of pressure sensitivities of flexible LC pressure sensors obtained from simulation and experiment.

Flexible LC Pressure Sensors	Pressure Sensitivity (Experimental Results)	Pressure Sensitivity (Finite Element Simulation)
The sensor based on Ecoflex material	−2.2 MHz/kPa	−1.5 MHz/kPa
The sensor based on PDMS material	−0.6 MHz/kPa	−0.76 MHz/kPa
The sensor based on thin PI substrate and Ecoflex micro-column dielectric	−2.4 MHz/kPa	

**Table 2 micromachines-12-00976-t002:** Comparison of pressure sensitivities of flexible LC pressure sensors in different documents.

Sensor	Sensitivity	Sensor	Sensitivity
Sensor in this paper	−2.2 MHz/kPa	Piezo Inkjet printing LC pressure sensor [22]	−0.083 MHz/kPa
Ultra-small flexible LC pressure sensor [16]	−10.7 MHz/kPa	Bioabsorbable wireless pressure sensor [23]	−1.5 MHz/kPa
LC pressure sensor based on textile [19]	−0.19 MHz/kPa	Wireless bioabsorbable pressure sensor embedded in orthopedic implants [24]	−0.045 MHz/kPa
LC pressure sensor for wound monitoring [18]	−2.2 MHz/kPa	Dual-parameter LC pressure sensor simultaneously monitors pressure and humidity [20]	−0.36 MHz/kPa
LC pressure sensor based on graphene/PDMS sponge [17]	−2.2 MHz/kPa	Thin film flexible LC pressure sensor [29]	−0.49 MHz/kPa

**Table 3 micromachines-12-00976-t003:** Comparison of the readout ability between the planar monopole antenna and the copper ring.

Readout Device	Vertical Distance	Horizontal Displacement	Rotation Angle
Planar monopole antenna	4 cm	8 cm	90°
Copper ring	2.5 cm	1.5 cm	90°

## Data Availability

The data that support the findings of this study are available from the corresponding author upon reasonable request.

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
