# Peer review of "Flexible Wireless Passive LC Pressure Sensor with Design Methodology and Cost-Effective Preparation"

_micromachines, 2021, doi:10.3390/mi12080976_

Round 1

Reviewer 1 Report

Dear authors,

Please update the manuscript with the comments suggested below to improve the quality of the manuscript.

  1. The abstract needs to be incorporated with the gist of the complete study in the manuscript. In the current version of the abstract, only the summary of the work is mentioned in a very broad view. Instead, please include specific details of the research study presented in the manuscript.
  2. In the abstract, please include the limitations of the current/previous technologies performed by the peers in this specific study: flexible and wireless pressure sensors. Also, please specify the scientific advances made in the proposed ‘Low-Cost Wireless flexible Passive Pressure Sensor, to overcome those limitations.
  3. Please update the abstract with the specific experimental conditions and operational parameters used in the evaluation of the performance of the flexible LC pressure sensor.
  4. In the introduction, please include more details of the peer studies performed on the ‘flexible and wireless pressure sensors, to guide the reader to understand the importance of the research study performed. Also, include corresponding references in the text when mentioning the details.
  5. In the introduction, please include the knowledge gaps existing in the current research work and prior studies performed in the field. Very importantly, please specify the need for the current work presented in the manuscript.
  6. In the last paragraph of the introduction, kindly include the details of the broader impacts on the study made and the results achieved. It is very important to provide the future scope of the research performed to make a strong impact on the readers on the research performed/Study proposed.
  7. In section 2.2: please incorporate the details of the comparison of the pressure sensitivity of various flexible LC pressure sensors. Also please update figure- 5e with the high-quality image as the current figure is not clear.
  8. In section-2, kindly incorporate the logical reasoning and scientific conclusions made from the plots in figures 7 & 8, also please use the ongoing research results of peers with appropriate references to support your arguments and statements.
  9. Please arrange the section of ‘conclusions’ to section-4, instead of section-3. Also please move the section-4: methods to section-2 and Section -2: Results and Discussion to Section-3. Kindly follow the below link for detailed instructions to prepare the manuscript and submit it. https://www.mdpi.com/journal/micromachines/instructions
  10. Please revise the manuscript with English grammar. There are many places that the manuscript needs to be improved with respect to English writing.

Reviewer 2 Report

In this paper, a sandwich-structured flexible LC pressure sensor based on Ecoflex micro-column as a dielectric is proposed. The flexible LC pressure sensor can be integrated into a sensing array and can be used to measure various physical activity signals for continuous health monitoring.

The manuscript is clear and well written. Results both from numerical simulations and measurements are presented and extensively commented. They demonstrate the good performance of the device since the resonance frequency of the sensor changes as a linear function of the applied pressure and has a higher sensitivity than most sensors reported in other existing works.

In addition, a simple and cheap method to fabricate the pressure sensor is presented. Even if I’m not an expert in manufacturing processes, the motivations exposed by the authors seem to be very convincing.

I have a question regarding the numerical simulations. While the sensitivity analysis carried out considering the variation of the different positions of the probe is well described, it is not clear to me what the optimization of the device consists of, other than the choice of the materials.
Have the authors also performed some sort of optimization of the geometric dimensions of the flexible LC pressure sensor?
Otherwise, for this scope, some pieces of information about the computing time and resources required by numerical simulations could be useful to the readers.

Another question is related to the computation of the effective capacitance using eq. (2). In this calculation, is it overlooked that the two plates are connected by the U-shaped conductor?

Finally, some minor remarks:

  1. in fig.1 the thickness of the Ecoflex substrate is missing;
  2. page 9 line 275, Fig.4(a) should be Fig.4(c);
  3. page 14 line 422, ref. [19] seems to be superscript.

Round 2

Reviewer 1 Report

Dear authors,
Thank you for updating the manuscript with recommended changes.